# Coordinated Localization and Antagonistic Function of NtPLC3 and PI4P 5-Kinases in the Subapical Plasma Membrane of Tobacco Pollen Tubes

**DOI:** 10.3390/plants9040452

**Published:** 2020-04-03

**Authors:** Irene Stenzel, Till Ischebeck, Linh Hai Vu-Becker, Mara Riechmann, Praveen Krishnamoorthy, Marta Fratini, Ingo Heilmann

**Affiliations:** 1Department of Cellular Biochemistry, Institute for Biochemistry and Biotechnology, Martin Luther University Halle-Wittenberg, 06120 Halle (Saale), Germany; irene.stenzel@biochemtech.uni-halle.de (I.S.); linhvu@gmx.de (L.H.V.-B.); mriechmann@biochem.uni-kiel.de (M.R.); praveen.krishnamoorthy@wustl.edu (P.K.); marta.fratini@bct.uni-halle.de (M.F.); 2Department of Plant Biochemistry, Albrecht-von-Haller-Institute of Plant Sciences and Göttingen Center for Molecular Biosciences (GZMB), Georg August University Göttingen, 37077 Göttingen, Germany; tischeb@gwdg.de

**Keywords:** Phosphatidylinositol 4,5-bisphosphate, PtdIns(4,5)P_2_, phospholipase C, PI4P 5-kinase, pollen tube tip swelling

## Abstract

Polar tip growth of pollen tubes is regulated by the membrane phospholipid phosphatidylinositol 4,5-bisphosphate (PtdIns(4,5)P_2_), which localizes in a well-defined region of the subapical plasma membrane. How the PtdIns(4,5)P_2_ region is maintained is currently unclear. In principle, the formation of PtdIns(4,5)P_2_ by PI4P 5-kinases can be counteracted by phospholipase C (PLC), which hydrolyzes PtdIns(4,5)P_2_. Here, we show that fluorescence-tagged tobacco NtPLC3 displays a subapical plasma membrane distribution which frames that of fluorescence-tagged PI4P 5-kinases, suggesting that NtPLC3 may modulate PtdIns(4,5)P_2_-mediated processes in pollen tubes. The expression of a dominant negative NtPLC3 variant resulted in pollen tube tip swelling, consistent with a delimiting effect on PtdIns(4,5)P_2_ production. When pollen tube morphologies were assessed as a quantitative read-out for PtdIns(4,5)P_2_ function, NtPLC3 reverted the effects of a coexpressed PI4P 5-kinase, demonstrating that NtPLC3-mediated breakdown of PtdIns(4,5)P_2_ antagonizes the effects of PtdIns(4,5)P_2_ overproduction in vivo. When analyzed by spinning disc microscopy, fluorescence-tagged NtPLC3 displayed discontinuous membrane distribution omitting punctate areas of the membrane, suggesting that NtPLC3 is involved in the spatial restriction of plasma membrane domains also at the nanodomain scale. Together, the data indicate that NtPLC3 may contribute to the spatial restriction of PtdIns(4,5)P_2_ in the subapical plasma membrane of pollen tubes.

## 1. Introduction

Pollen tubes are tip-growing cells with a key function in plant sexual reproduction [1]. The elongation of pollen tubes is strictly limited to the cells’ apex, from where cell wall material is secreted [2,3,4,5]. In the tip of the growing pollen tube, secretion, membrane trafficking and cytoskeletal elements are controlled in part by the membrane phospholipid phosphatidylinositol 4,5-bisphosphate (PtdIns(4,5)P_2_) [3,6]. To exert its regulatory functions, PtdIns(4,5)P_2_ binds to cytoplasmic proteins possessing specific lipid binding domains, which are recruited to the plasma membrane by their lipid ligand [7,8]. The recruitment of PtdIns(4,5)P_2_-binding proteins is thought to underlie the dynamic plasma membrane association of the protein complexes required for cell expansion [3,9,10]. 

PtdIns(4,5)P_2_ is formed by PI4P 5-kinases, which are encoded in the Arabidopsis genome by a family of eleven genes [11]. Based on their domain structure, PI4P 5-kinases can be categorized in subfamilies A (isoforms PIP5K10 and PIP5K11) and B (isoforms PIP5K1-PIP5K9), and the isoforms PIP5K2, PIP4K4, PIP5K5, PIP5K6, PIP5K10 and PIP5K11 are expressed in pollen and pollen tubes [7,8]. The importance of PtdIns(4,5)P_2_ formation for pollen tube growth is underlined by the observation that Arabidopsis mutants with defects in genes encoding PI4P 5-kinases display reduced pollen germination, and diminished pollen tube growth for cells that succeed in emerging, including *pip5k4 pip5k5* double mutants and *pip5k10 pip5k11* double mutants [12,13,14,15]. 

In line with its important function in controlling polar tip-growth, fluorescent probes for PtdIns(4,5)P_2_ indicate an enrichment of this lipid in a subapical plasma membrane region of pollen tubes which extends from 2-3 µm to approximately 25 µm below the pollen tube apex [12,13,14,15,16,17]. The distribution pattern of the PtdIns(4,5)P_2_ biosensors is largely mirrored by the distribution of fluorescence-tagged PI4P 5-kinases in these cells, and it has been proposed that the spatial positioning of PI4P 5-kinases is a factor controlling the extent of PtdIns(4,5)P_2_ in the plasma membrane of pollen tubes [12,13,18]. The spatial dimensions of the PtdIns(4,5)P_2_ domain in the subapical plasma membrane are critical for polar tip-growth, as evidenced by the observation that the overproduction of PtdIns(4,5)P_2_, upon overexpression of PI4P 5-kinases, and the ensuing enlargement of the plasma membrane domain occupied by this lipid results in altered patterns of pollen tube expansion [12,13]. A loss of apical cell polarity upon overproduction of PtdIns(4,5)P_2_ gives rise to substantial changes in pollen tube cell shape, including apical tip swelling [13,19] or in more complex cases, branched cell morphologies [12,14,16,19]. From these observations, it is clear that the extent of the subapical PtdIns(4,5)P_2_ domain must be tightly controlled to enable "normal" pollen tube growth.

Besides the biosynthesis of PtdIns(4,5)P_2_, its degradation can also contribute to defining its spatial distribution in the apical plasma membrane of pollen tubes. PtdIns(4,5)P_2_ can be hydrolyzed by phospholipase C (PLC) to form diacylglycerol (DAG) and inositol 1,4,5-trisphosphate (InsP_3_), thus removing PtdIns(4,5)P_2_ as a polarized recruitment signal from the plasma membrane [7,8]. The Arabidopsis genome encodes nine isoforms of PLC [8,11], four of which are expressed in pollen. So far, the role for PLCs in Arabidopsis pollen tube growth has not been analyzed in detail. In previous experiments, it was shown that a fluorescence-tagged variant of the tobacco (*Nicotiana tabacum*) PLC, NtPLC3, displays a subapical plasma membrane distribution in tobacco pollen tubes [20], which at first approximation appears similar to that observed for PI4P 5-kinases. A fluorescence-tagged variant of PLC1 from petunia *(Petunia inflata)* displayed an equivalent localization pattern when expressed in petunia pollen tubes [21]. Both NtPLC3 and PLC1 from petunia are active phosphoinositide-specific PLCs and most similar to the PLCζ family [22]. Interestingly, the expression of a dominant negative variant of petunia PLC1 resulted in substantial pollen tube tip swelling [21], consistent with delimited PtdIns(4,5)P_2_ formation upon reduced hydrolysis by the catalytically inactive PLC1 from petunia. 

While the localization and cell morphology data suggest that PLCs may serve as PtdIns(4,5)P_2_-degrading enzymes in pollen tubes and functionally antagonize PtdIns(4,5)P_2_ formation by PI4P 5-kinases, there has been no dedicated analysis of the functional interplay of PLCs and PI4P 5-kinases to date. The pollen tube system is an ideal model to study the functional interrelations of PLC with PI4P 5-kinase because of the well-defined subapical plasma membrane domain, which is easy to monitor by confocal microscopy, and because of the well-characterized cell morphologies ensuing from the overproduction of PtdIns(4,5)P_2_.

Here, we demonstrate that fluorescence-tagged NtPLC3 displays a subapical plasma membrane distribution which encompasses a slightly narrower plasma membrane region decorated by the PI4P 5-kinases PIP5K2 or PIP5K11. A catalytically inactive variant of NtPLC3 exerts a substantial dominant negative effect when expressed in pollen tubes, which closely reflects the effect of overexpressed PIP5K2. In coexpression experiments, NtPLC3 functionally antagonizes the effects of overexpressed PIP5K2, resulting in a large proportion of normal growing pollen tubes upon coexpression of both enzymes. Together with a plasma membrane distribution in a discontinuous pattern omitting punctate areas determined by spinning disc (SD) analysis, our data suggest that NtPLC3 may contribute to the spatial restriction of PtdIns(4,5)P_2_ in the plasma membrane of pollen tubes.

## 2. Results

PtdIns(4,5)P_2_ in the subapical plasma membrane domain of pollen tubes is formed by PI4P 5-kinases and can be degraded by PLC. Here, we address the interplay of these enzymes to balance PtdIns(4,5)P_2_ biosynthesis and degradation

### 2.1. NtPLC3 Localizes to A Subapical Membrane Domain Encompassing Domains Occupied by PIP5K2 or PIP5K11. 

It has previously been reported that the distribution of PtdIns(4,5)P_2_ in pollen tubes, as indicated by fluorescent biosensors such as RedStar-PLCδ1-PH, closely matches the distribution of PI4P 5-kinases [12,13]; a similar pattern was also reported for fungal hyphae [23]. The relative extent of the subapical plasma membrane regions decorated by RedStar-PLCδ1-PH or by PIP5K11 was verified for tobacco pollen tubes in coexpression experiments (Figure 1A). To prevent morphological changes which PIP5K11 overexpression might cause, expression levels were kept low. The observation that the plasma membrane region marked by the PtdIns(4,5)P_2_ biosensor does not extend beyond its site of biosynthesis suggested that the lateral diffusion of PtdIns(4,5)P_2_ might be limited, possibly by a PtdIns(4,5)P_2_-degrading enzyme such as PLC. Therefore, the relative spatial distributions of PI4P 5-kinases and PLC in the subapical plasma membrane region of tobacco pollen tubes was analyzed in coexpression experiments (Figure 1B–E). The previously reported NtPLC3 was selected for this analysis because it is well characterized and has been shown to occupy a subapical plasma membrane domain in tobacco pollen tubes [20]. This pattern was compared to the distribution of the respectively coexpressed PI4P 5-kinases PIP5K2 [19,24,25,26] and PIP5K11 [13], which represent the two subfamilies, i.e., A and B, of Arabidopsis PI4P 5-kinases [11]. When a PIP5K2-mCherry fusion was coexpressed at low levels with EYFP-NtPLC3, both enzymes decorated subapical plasma membrane domains (Figure 1B), with the noted distinction that the membrane domain occupied by EYFP-NtPLC3 was elongated over the subapical membrane region and spanned the entire region occupied by AtPIP5K2-mCherry. PIP5K2-mCherry localized more to the pollen tube tip with some signal in the subapical region, whereas EYFP-NtPLC3 localized mainly to the pollen tube shank. When EYFP-PIP5K11 was coexpressed with NtPLC3-RFP, the NtPLC-RFP marker spanned a larger plasma membrane domain encompassing the region decorated by EYFP-PIP5K11 (Figure 1C) in a pattern resembling that observed for PIP5K2-mCherry and EYFP-NtPLC3 (Figure 1A). The dimensions of the subapical plasma membrane regions occupied by NtPLC3-RFP or EYFP-PIP5K11 were monitored over time, as indicated by the durations in the individual panels of selected frames (Figure 1C). A two-channel kymograph analysis over a period of 10 min at a frame rate of ~2 frames per minute indicated that the distance of the occupied areas from the growing pollen tube tip remained roughly constant (Figure 1D). The fast growth rate and nonlinear growth of the pollen tubes required manual adjustment to obtain the kymographs. To provide an independent analysis of the relative distribution patterns of NtPLC3 and PIP5K11, the intensity and distribution of the PIP5K11 and PLC signals were additionally analyzed using the machine learning program Ilastik [27], minimizing bias (Figure 1E)**.** The resulting very similar pattern suggests that NtPLC3 continuously accompanies PI4P 5-kinases in the subapical plasma membrane and towards the shank of the pollen tube, thereby possibly preventing the uncontrolled diffusion of PtdIns(4,5)P_2_ far from its sites of biosynthesis [28].

### 2.2. Expression of Dominant Negative NtPLC3-RFP H124A Results in Pollen Tube Tip Swelling.

To test the functional effects of NtPLC3 on pollen tube growth, we created a dominant negative variant, NtPLC3 H124A, in analogy to the previous report by Dowd and co-workers [21]. Both NtPLC3 and NtPLC3 H124A were recombinantly expressed in *Escherichia coli* as fusions to an N-terminal maltose binding protein (MBP) tag for enhanced solubility and tested for activity (Figure 2). PLC activity was determined in vitro by monitoring the release of radiolabel from a hydrophobic 2[^3^H]PtdIns(4,5)P_2_ substrate from an organic into an aqueous solvent phase, as previously shown [29,30]. When the recombinant proteins (Figure 2A) were tested in vitro, the catalytic activity of MBP-NtPLC3 H124A protein was reduced by 99 % compared to wild type MBP-NtPLC3 protein, and did not differ from that observed with the MBP negative control (Figure 2B). The biochemical tests enabled the use of the cataly tically inactive NtPLC3 H124A as a dominant negative variant of NtPLC3.

While experimental conditions for the localization studies shown in Figure 1 were chosen to ensure weak expression and normal cell growth, constructs were now intentionally overexpressed to test their effects on cell morphology. Overexpression of the dominant negative NtPLC3-RFP H124A in tobacco pollen tubes resulted in pollen tube tip swelling (Figure 3A), similar to the pattern previously reported for the expression of dominant negative Petunia PLC1 [21]. We assume that the overexpression of NtPLC-RFP H124A from the Lat52 promoter was strong enough to exert the dominant negative effect by displacing a substantial proportion of the intrinsic PLC3 population present in the pollen tube cells. The expression of EYFP or active NtPLC3 served as negative controls and resulted in pollen tube morphologies similar to those of untransformed pollen tubes (Figure 3A). The overexpression of NtPLC3-RFP H124A resulted in pollen tubes with tip diameters of up to 27 µm (Figure 3B). This phenotypic effect was similar, although to a slightly lesser extent, to cell morphologies observed upon overexpression of PI4P 5-kinases, such as PIP5K2 or PIP5K11 (Figure 3B). No pollen tube tip swelling was observed upon overexpression of PIP5K5 (Figure 3A and B), which is consistent with previous reports stating that the expression of this enzyme results in overactive apical secretion and a tip branching phenotype [12,14]. To further characterize the effect of NtPLC3, the incidence of pollen tube morphologies previously observed upon overproduction of PtdIns(4,5)P_2_ was scored, including the analysis of tip branching and tip swelling (Figure 3C). 

Interestingly, while pollen tube tip swelling was observed in ~60% of cells expressing the dominant negative NtPLC-RFP H124A (Figure 3C), in a pattern similar to previous observations on petunia pollen tubes [21], no tip branching was observed. Tip branching was positively observed in our experiments upon overexpression of PIP5K5, indicating that branching could occur under the experimental conditions used. Differences in pollen tube morphologies between samples were not the result of different expression levels of the expressed proteins, as fluorescence intensities were similar (Figure 3D). While the intensity measurements indicate mean values for the pollen tube tip region, we observed enhanced cytoplasmic fluorescence in pollen tubes displaying tip swelling, as, for instance, upon overexpression of NtPLC3-RFP H124 A (Figure 3A). Considering further previous observations on tip swelling upon PtdIns(4,5)P_2_ overproduction [19,31], the effects of the expression of dominant negative NtPLC3-RFP H124A are consistent with increased formation of PtdIns(4,5)P_2_, and suggest that NtPLC3 might antagonize PtdIns(4,5)P_2_ production mediated by PI4P 5-kinases.

### 2.3. NtPLC3-RFP Antagonizes the Effects of Coexpressed PIP5K2-EYFP on Pollen Tube Tip Swelling.

To elucidate the functional interplay of NtPLC3 with PIP5K2, we coexpressed AtPIP5K2-EYFP with either NtPLC3-RFP or NtPLC3-RFP H124A and quantified the ensuing effects on pollen tube morphologies (Figure 4). In these experiments, the overexpression of AtPIP5K2-EYFP resulted in ~75% of cells displaying tip swelling (Figure 4A). This effect was abated upon coexpression of AtPIP5K2-EYFP with NtPLC3-RFP, but not upon coexpression with NtPLC3-RFP H124A (Figure 4A), indicating that NtPLC3-RFP counteracted the effects of overexpressed AtPIP5K2-EYFP. Coexpression of AtPIP5K2-EYFP with NtPLC3-RFP H124A resulted in a slight increase in tip swelling (Figure 4A), but no fully additive effect, suggesting that AtPIP5K2-EYFP and NtPLC3-RFP H124A may modulate the same pool of PtdIns(4,5)P_2_. No combination of the expressed proteins resulted in detectable pollen tube tip branching, and no substantial differences in the expression levels of the different proteins were observed in the transformed pollen tubes, based on the detected fluorescence intensities (Figure 4B). Together, the results support the hypothesis that NtPLC3 expression has an antagonistic effect on the pollen tube tip-swelling phenotype arising from AtPIP5K2 activity.

### 2.4. NtPLC3-EYFP Displays Discontinuous Plasma Membrane Localization Omitting Circular Nanodomains. 

Our results suggested that NtPLC3 contributes to the modulation of the PtdIns(4,5)P_2_ domain in the subapical plasma membrane of pollen tubes. This interpretation is consistent with the initial observation that NtPLC3 decorates an expanded subapical plasma membrane domain compared to PIP5K2 or PIP5K11 (Figure 1). Therefore, the distribution of NtPLC3-EYFP in the subapical plasma membrane of pollen tubes was analyzed in more detail by SD microscopy. When observed by SD microscopy, the NtPLC3-EYFP marker displayed uniform localization throughout the plasma membrane, with the noted exception that fluorescence spared out certain dot-like membrane areas (Figure 5A, B). Intensity profiles such as that recorded along the dashed line in Figure 5B indicate that the plasma membrane fluorescence of NtPLC3-EYFP dropped sharply in the dark dots from gray values of ~200 units to values below ~75 units (Figure 5B). The dark dots with reduced NtPLC3-EYFP fluorescence had diameters of ~0.5 µm (Figure 5C) and appeared at a density of ~0.5 ± 0.2 µm^−2^ (Figure 5D). The diameter of the dark dots and their area density were analyzed with increasing distance to the pollen tube tip over the region spanned by NtPLC3. While dot diameters did not change (Figure 5C), there was a minor but significant reduction in dot density with increasing distance from the tip (Figure 5D). A control experiment was performed by analyzing the plasma membrane distribution of PIP5K5-EYFP (Figure 5E, F). PIP5K5-EYFP also localized to the subapical plasma membrane of pollen tubes, but when analyzed by SD microscopy, displayed a uniform distribution in the plane of the plasma membrane (Figure 5F). Overall, the data suggest that NtPLC3 might influence PtdIns(4,5)P_2_ at the nanodomain scale.

## 3. Discussion

The dynamic control of PtdIns(4,5)P_2_ production in the subapical plasma membrane of pollen tubes is critical for polar cell expansion. In this study, we addressed how pollen tube growth is influenced by the contrapuntal activities of PI4P 5-kinase and PLC, which act in biosynthesis and breakdown of PtdIns(4,5)P_2_, respectively. Our data indicate that NtPLC3 can functionally antagonize the effects of PtdIns(4,5)P_2_ formed by PI4P 5-kinases in vivo. This notion is supported by the observations that i) NtPLC3 localizes in a plasma membrane region encompassing that which is occupied by PIP5K2 (Figure 1); ii) the expression of the dominant negative NtPLC3 H124A resulted in similar tip swelling of pollen tubes as the expression of the PI4P 5-kinases PIP5K2 or PIP5K11 (Figure 3); and iii) the expression of NtPLC3, but not of its inactive variant NtPLC3 H124A, abated the effects of the co-overexpression of PIP5K2 on pollen tube morphology (Figure 4). Our SD-based analysis furthermore shows an unexpected fluorescence distribution of NtPLC3 in the subapical plasma membrane of pollen tubes, where the enzyme occupies the majority of the membrane plane but is excluded from small punctate patches resembling plasma membrane "nanodomains" (Figure 5). Together, these findings suggest that NtPLC3 functionally antagonizes PI4P 5-kinases in vivo to modulate PtdIns(4,5)P_2_ with a key role in the control of pollen tube growth.

It should be noted that the fluorescence patterns or physiological effects observed upon overexpression of PI4P 5-kinases or other enzymes in pollen tubes under the control of strong pollen-specific promoters, such as Lat52 [32], may not reflect the situation in unperturbed cells. However, transient expression by particle bombardment can be used to obtain a range of expression intensities, which, for PI4P 5-kinases, were previously be found to correlate with the degree of ensuing morphological alterations of the pollen tube cells. In morphologically normal cells, subcellular distribution patterns obtained for different fluorescence-tagged PI4P 5-kinases by transient Lat52-driven expression have previously been shown to faithfully reflect the patterns found for these fusion proteins in Arabidopsis pollen tubes when expressed from their respective intrinsic promoters in the corresponding Arabidopsis mutant backgrounds [12,13].

Based on the fluorescence distribution of NtPLC3 observed by SD (Figure 5), it is tempting to speculate whether the punctate plasma membrane areas not occupied by NtPLC3 might be sites of enhanced PtdIns(4,5)P_2_ formation and/or residence of PI4P 5-kinases. In support of this notion, the presence of PtdIns(4,5)P_2_ nanodomains has previously been proposed, based on the analysis of plant plasma membrane preparations by transmission electron microscopy and immuno gold labeling [33]. Furthermore, small GTPases of the ROP family, which are important for pollen tube growth [10] and are functionally linked to PtdIns(4,5)P_2_ [13,17], have been observed to reside in membrane nanodomains [34]. PtdIns(4,5)P_2_ may be restricted to nanodomains by the action of degrading enzymes, such as PLC, as previously proposed [7]. Our model (Figure 6) illustrates a possible role for NtPLC3 in restricting PtdIns(4,5)P_2_ formation in some areas while enabling PtdIns(4,5)P_2_ formation in others. 

While this model attempts to reflect the data without much speculation, the molecular basis for the interesting localization pattern of NtPLC3 remains currently unclear. It is possible that local asymmetries in the distribution of certain membrane lipids, such as sphingolipids or sterols, and the ensuing differences in the biophysical properties of certain membrane areas, may be important aspects regarding the correct recruitment of NtPLC3 to regions where its activity is required [35,36,37,38]. Unfortunately, the analysis of such interrelations by fluorescence microscopy is impeded by the limitations of the biosensors and fluorescent markers that are currently available, which exert a substantial influence on the function of the intrinsic phosphoinositide system in vivo [8,39,40]. With respect to the actual study, the expression of NtPLC3, an active enzyme degrading PtdIns(4,5)P_2_, will influence the distribution of PtdIns(4,5)P_2_ biosensors. Reciprocally, the binding of a PtdIns(4,5)P_2_ biosensor to its lipid target might influence the accessibility of the lipid for degradation by PLC [41]. Possibly, for these or related reasons, so far we have not been able to observe meaningful localization patterns of NtPLC3-RFP vs. a coexpressed biosensor for PtdIns(4,5)P_2_. It is possible that future advances in biosensor development or in imaging technology may help us to circumvent some of these experimental caveats.

At a larger scale, our study demonstrates antagonistic effects of PI4P 5-kinases and PLC on a relevant physiological process in vivo, and the proposed functional interplay can easily be rationalized from the known in vitro activities of these enzymes. In addition, the data hold a number of unexpected findings, which will require further study in the future to complete the picture. For instance, it remains unclear why the expression of the dominant negative NtPLC3 H124A resulted in tip swelling, but not in any detectable pollen tube tip branching. Both pollen tube tip swelling [13,19] and tip branching [12,14,19] have previously been shown to result from the overproduction of PtdIns(4,5)P_2_ through different isoforms of PI4P 5-kinases. Our data on the effects of expressed NtPLC3 H124A are consistent with the corresponding results on petunia PLC1, where also only pollen tube tip swelling was observed [21].

Overall, it appears possible that NtPLC3 or petunia PLC1 can antagonize some but not other regulatory pathways mediated by PtdIns(4,5)P_2_, possibly as a consequence of a difference in the accessibility or subcompartmentalization of PtdIns(4,5)P_2_ formed by different PI4P 5-kinase isoforms in the pollen tube plasma membrane. This notion is consistent with previous results on the functionality of hypervariable linker domains of plant PI4P 5-kinases in mediating membrane recruitment into different regulatory contexts [19]. The effect of PtdIns(4,5)P_2_ on pollen tube tip swelling has previously been rationalized through a stabilizing effect of PtdIns(4,5)P_2_ on the actin cytoskeleton [13]. It remains to be seen whether NtPLC3 might indirectly also influence the dynamics of the actin cytoskeleton in pollen tubes through its effects on PtdIns(4,5)P_2_. Future experiments will elucidate whether the antagonistic effects of PLC with PI4P 5-kinases may be relevant for pollen tube functionality, genetic transmission or seed set.

## 4. Materials and Methods 

### 4.1. cDNA Constructs

The coding sequence for NtPLC3 was amplified from tobacco cDNA using the following primer combination 5’-GATCCCATGGCATCGAGACAGACGTACAGAGTCT-3’/5’-GATCCCATGGACATATTCGAAACGCATAAGAAGC-3’, and moved as an NcoI/NcoI fragment into the expression plasmid pET M-41, encoding the NtPLC3 protein with an N-terminal maltose binding protein (MBP) tag. Site-directed mutagenesis was performed on the NtPLC3 coding sequence using the primer combination 5’-TCTCATTACTTCATATACACAGGAGCTAATTCCTATCTAACTGGGAATCAA-3’/ 5’-TTGATTCCCAGTTAGATAGGAATTAGCTCCTGTGTATATGAAGTAATGAGA-3’, yielding NtPLC3 H124A. For expression as fluorescent fusions, the coding sequences for NtPLC3 or NtPLC3 H124A were subcloned into pEntryD and moved by Gateway technology into the pLat52-GW vector [12] in frame with the coding sequence for RFP. Constructs for the expression of fluorescent fusions of Arabidopsis PIP5K2 [19] or PIP5K11 [13] were used as previously reported.

### 4.2. Recombinant Protein Expression

Protein extracts containing recombinant NtPLC3 and NtPLC3 H124A protein were generated by the expression in *E. coli* Rosetta 2 cells. Cells were grown in liquid LB media, selecting for kanamycin and chloramphenicol resistance. Expression cultures of 100 ml were incubated at 37 °C with shaking at 200 rpm. Expression was induced at an optical density of 0.8 by adding 1 mM isopropyl-β-D-thiogalactosylpyranoside (IPTG). Cultures were incubated overnight at 25 °C with shaking at 200 rpm and harvested by centrifugation. The cell sediment was resuspended on ice in lysis buffer containing 50 mM Tris-HCl, pH 8, 300 mM NaCl, 1 mM EDTA, 10 % (*w/v*) glycerol and 50 U of lysozyme. Cells were ruptured by ultrasound on ice, using five 60 s bursts in a Sonifier Cell Disruptor B15 (Branson Dietzenbach, Germany) at 50% power and 50% impulse settings. The protein extracts were cleared by centrifugation and stored at −20 °C. Proteins were analyzed by immunodetection as previously described [42,43].

### 4.3. Assay for PLC Activity

PLC activity was assayed in vitro according to the release of radiolabel from the 2[^3^H]PtdIns(4,5)P_2_ substrate from an organic to an aqueous solvent phase, as previously described [29]. In brief, 2µCi of 2[^3^H]PtdIns(4,5)P_2_ substrate (DuPont/New England Nuclear, Wellesley, MA, USA) was mixed in chloroform with 8 µg of unlabeled PtdIns(4,5)P_2_ (Avanti, Alabaster, AL, USA) at a ratio of 2/8 (*v/v*) and dried in a glass reaction vial. The lipid coat was redissolved in 5 μL of 1 % (*w/v*) Triton X-100 and sonified on ice in a waterbath sonicator. Recombinant protein extract was added at 5–10 µg in a volume of 30 µL. Then, 20 µL of reaction buffer containing 50 mM Tris-HCl, pH 6.4, and 20 μM CaCl_2_ were added and the samples incubated at 25 °C for 10 min. Reactions were stopped by adding 500 μL of chloroform:methanol (2:1) (*v/v*) and 500 μL 1 M HCl. Phases were allowed to separate and the aqueous phase was collected. The radiolabel in the aqueous phase was analyzed with a liquid scintillation counter (LSA Tricarb 1900 TR, Canberra Packard, Dreieich, Germany).

### 4.4. Transient Expression of Constructs in Tobacco Pollen Tubes

Mature pollen grains were collected from four to five tobacco flowers of approximately eight- to nine-week-old plants. The pollens were resuspended in liquid pollen tube growth media [44], followed by filtering of the pollen grains onto a cellulose acetate filter, before being transferred onto Whatmann paper which was moistened with pollen tube growth media. They were immediately transformed by bombardment with plasmid-coated 1-μm gold particles with a helium-driven particle accelerator (PDS-1000/He; Bio-Rad) using 1350 psi rupture disks and a vacuum of 28 inches of mercury, as previously described [45]. Prior to the bombardment, gold particles were coated with 4–5 µg of the desired plasmid of interest. After bombardment, the pollen grains were transferred into 300 µL of pollen tube growth media which was then equally divided onto three microscopic glass slides and viewed under the microscope 4−6 h after the bombardment.

### 4.5. Live Cell Microscopy and Image Processing 

Images were acquired with confocal Zeiss LSM510 meta or Zeiss LSM880 Airyscan systems, or with a Zeiss Cell observer SD with a Yokogawa CSU-X1 SD unit. SD microscopy allowed us to capture confocal imaging information with short dwell-times by using a rotating disc to generate multiple pin-holes in parallel. Short dwell times are additionally enabled by using a camera, which is more light-sensitive than a confocal point scanner, and can capture fluorescence signals of low intensity. For imaging with the LSM880, a 20x or a 63x oil immersion objective was used and images were captured with the ZEN Black image analysis software. During acquisition with the LSM880, EYFP was excited at 514 nm laser line and imaged with a HFT 514 nm major beam splitter (MBS) while mCherry or RFP was excited with a 561nm laser line and imaged with a HFT 561nm MBS. Image acquisition with the Zeiss Cell observer SD was performed with a 63x oil immersion objective and captured with a Photometrics Evolve 512 Delta EM-CCD camera. YFP was excited at 491 nm and mCherry at 561 nm using a multichannel dichroic and an ET525/50M or an ET595/50M band pass emission filter (Chroma Technology) for GFP and mCherry respectively. The Zen Blue image analysis software was used for image processing. FM 4-64 was added to pollen tubes at a final concentration of 10 µM, as previously described [46], and visualized after 5–15 min of incubation. Kymographs were obtained using Fiji image analysis software [47] by plotting fluorescence intensities in the region of interest over time. Since pollen tubes exhibit a fast growth rate and their growth is not linear, it was difficult to track the fluorescence intensities values of PLC and PIP5K11 at the cell membrane without substantial manual adjustment. To eliminate possible bias and provide an independent analysis of the relative distribution patterns of PLC3 and PIP5K11, the machine learning program Ilastik was used [27] to analyze the intensity and distribution of the PIP5K11 and PLC signals. By using the pixel classification and object classification tool, one time frame was used to train the program. Afterwards, the batch processing function analyzed all the remaining frames with no further manual processing. 

### 4.6. Statistical Evaluation

All quantitative data were tested for statistical significance using two-tailed Student’s t tests. Confidence intervals are given in the figure legends for each data set.

### 4.7. Accession Numbers

Sequence data from this article can be found in the GenBank/EMBL data libraries under the following accession numbers: AtPIP5K2, At1g77740, AtPIP5K5, At2g41210, AtPIP5K11, At1g01460, NtPLC3 and EF043044

## Figures and Tables

**Figure 1 plants-09-00452-f001:**
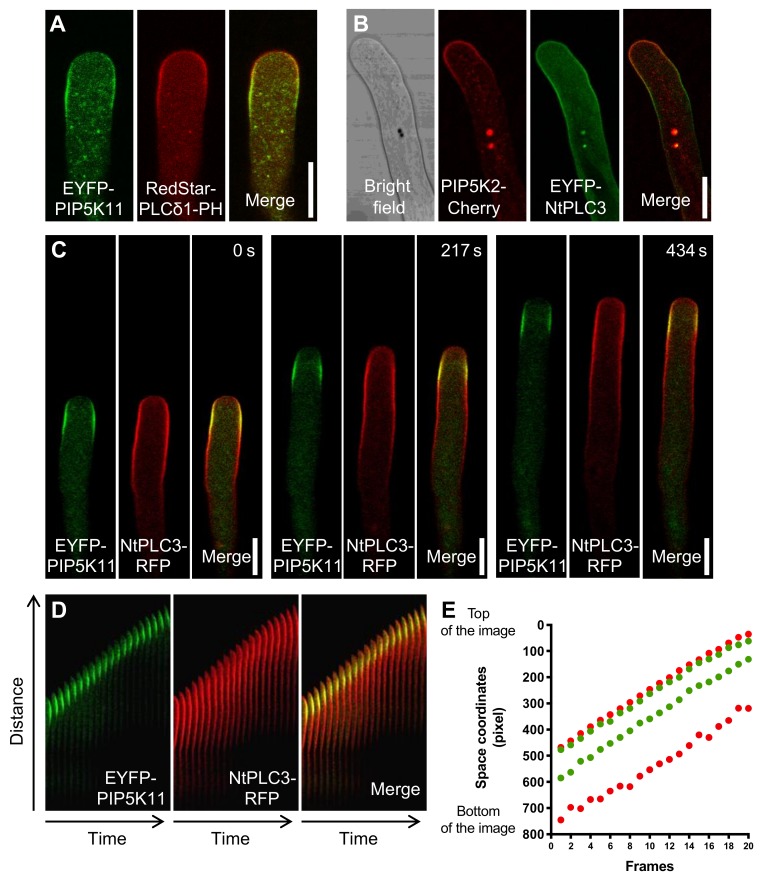
**NtPLC3 localizes to a subapical plasma membrane domain encompassing domains occupied by PIP5K2 or PIP5K11.** The localization of NtPLC3 was monitored by confocal imaging upon transient expression in tobacco pollen tubes. For these localization studies, expression levels were kept low to minimize morphological changes of the pollen tube cells. (**A**) Coexpression of EYFP-PIP5K11 with RedStar-PLCδ1-PH, a fluorescent biosensor for PtdIns(4,5)P_2_, as indicated. B-E, Association of fluorescence-tagged NtPLC3 with an extended subapical plasma membrane domain, which spans the region occupied by coexpressed PIP5K2-mCherry or EYFP-PIP5K11. (**B**) Coexpression of EYFP-NtPLC3 with PIP5K2-mCherry. (**C**) Coexpression of NtPLC3-RFP with EYFP-PIP5K11, monitored in a time lapse experiment. The time series shown was recorded for 600 s at a frame rate of 2.14 frames min^-1^. Three selected frames are shown, as indicated. (**D**) Kymograph analysis of median confocal LSM sections of a growing pollen tube from (C), indicating the dynamic relative movements of PIP5K2-EYFP (green) and NtPLC3-RFP (red). Right panel: merged images (overlap indicated by yellow). (**E**) Intensity and distribution of the EYFP-PIP5K11 (green) and NtPLC3-RFP (red) signals were further analyzed by the machine learning program Ilastik to provide nonbiased results. Note that the relative distances from the pollen tube tip are roughly constant, and that the region occupied by NtPLC3-RFP continuously spans the region occupied by PIP5K2-EYFP. Data are representative for five independent experiments. Scale bars = 10 µm.

**Figure 2 plants-09-00452-f002:**
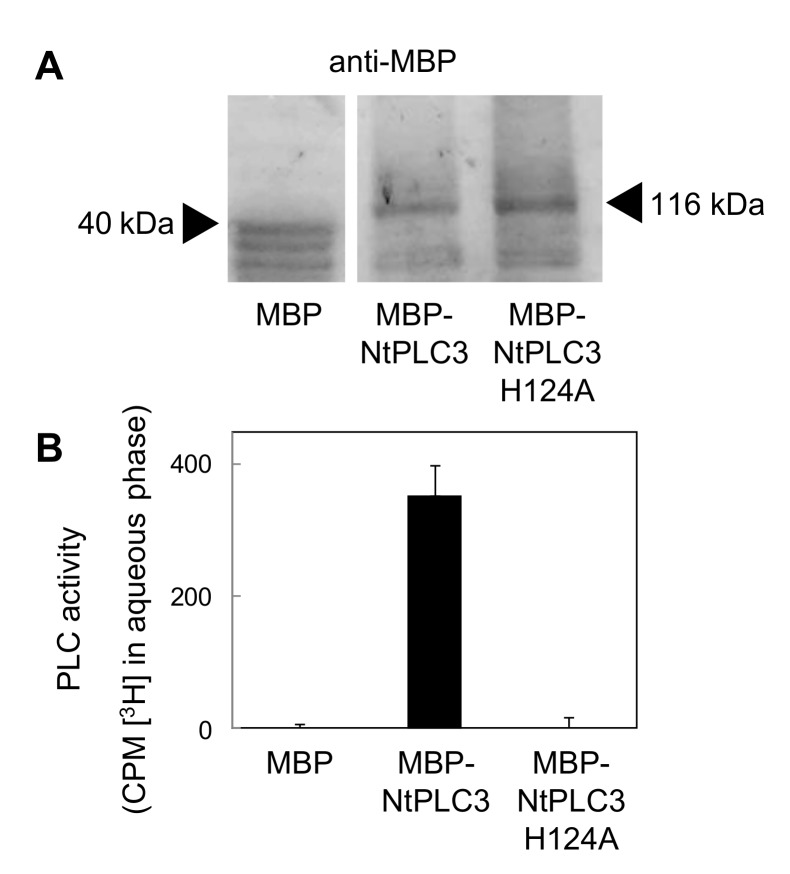
**A catalytically inactive NtPLC3 variant, NtPLC3 H124A.** A catalytically inactive variant of NtPLC3 to study dominant negative effects in vivo was generated by substituting histidine 124 for alanine. The effect of this substitution on enzyme function was tested using protein recombinantly produced in *E. coli* as a fusion to an N-terminal maltose-binding protein (MBP)-tag. (**A**) Immuno detection of recombinant MBP control, MBP-NtPLC3 and the dominant negative MBP-NtPLC3 H124A using an anti-MBP antibody, as indicated. (**B**) Catalytic activity of recombinant MBP, MBP-NtPLC3 and MBP-Nt-PLC3 H124A against a 2[^3^H]PtdIns(4,5)P_2_ substrate in vitro. Data are the mean ± SD from three experiments.

**Figure 3 plants-09-00452-f003:**
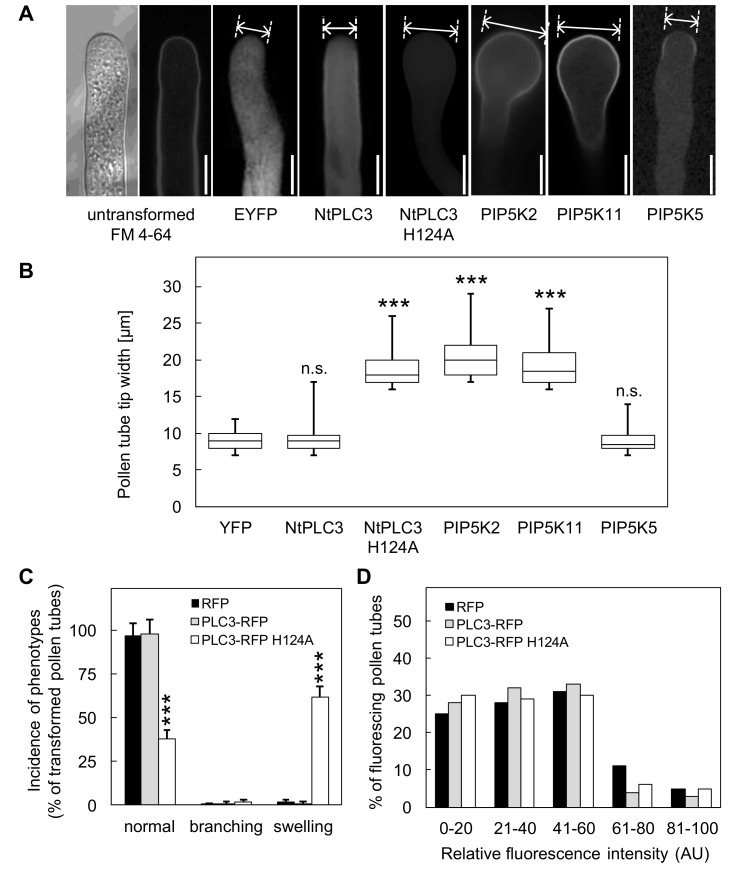
Overexpression of dominant negative RFP-NtPLC3 H124A induces pollen tube tip swelling, similar to expressed PI4P 5-kinases. The effects of NtPLC3 or NtPLC3 H124A on cell morphologies were tested upon transient overexpression in tobacco pollen tubes. The expression of all constructs was driven by the strong pollen-specific Lat52 promoter. (**A**) The extent of pollen tube tip swelling was assessed upon overexpression of NtPLC3, NtPLC3 H124A, PIP5K2, PIP5K11, or PIP5K5. Negative controls included untransformed pollen tubes, stained with the membrane dye, FM 4-64, or pollen tubes expressing EYFP, as indicated. (**B**) Pollen tube tip widths as determined for the respective expressed constructs. Asterisks indicate a significant difference to the YFP control according to a Student’s t-test (***, *p* ≤ 0.01); n.s., not significant. Data are from at least five indepen dent experiments and >50 cells were analyzed for each transformation. (**C**) Based on the same data set, the incidence of normal, branched or swollen cell morphologies was determined upon transient expression of RFP, NtPLC3-RFP or NtPLC3-RFP H124A in tobacco pollen tubes. Note that overex pression of the dominant negative NtPLC3-RFP H124A mediated pollen tube tip swelling, not tip branching. Asterisks indicate a significant difference to the RFP control according to a Student’s t-test (***, *p* ≤ 0.01). Data are from at least five independent experiments, and >50 cells were analyzed for each transformation. (**D**) Fluorescence intensities recorded during the expression of RFP, NtPLC3-RFP or NtPLC3-RFP H124A. Scale bars, 10 µm.

**Figure 4 plants-09-00452-f004:**
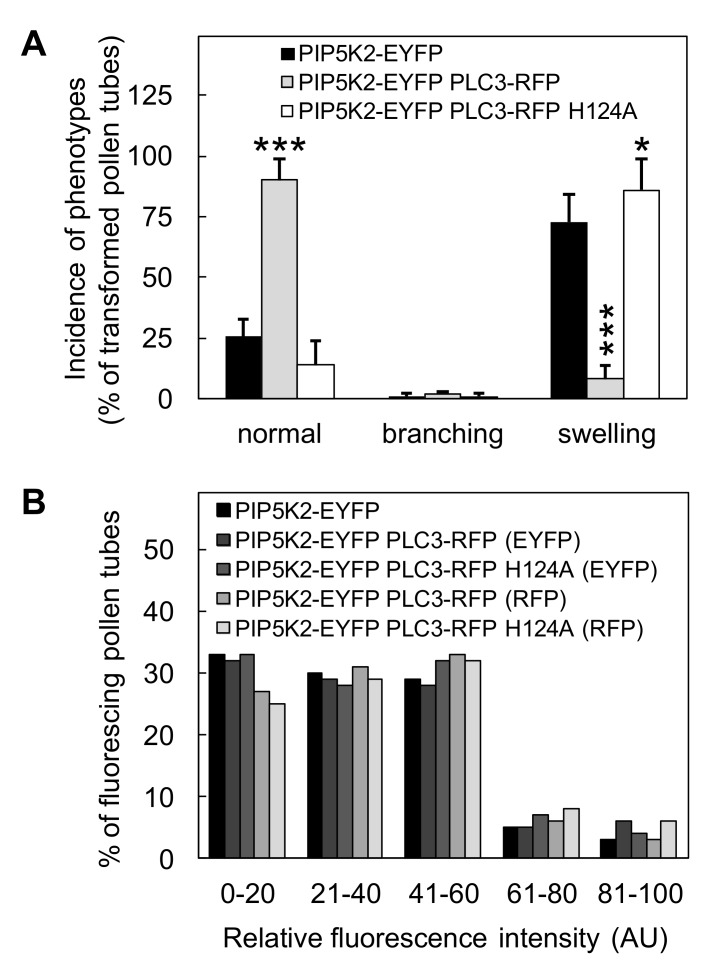
**NtPLC3-RFP counteracts PIP5K2-EYFP-induced PtdIns(4,5)P_2_-dependent tip swelling.** The effects of NtPLC3 on cell morphologies mediated by the overexpression of PIP5K2 were tested upon transient co-overexpression in tobacco pollen tubes. The expression of all constructs was driven by the strong pollen-specific Lat52 promoter. (**A**) Incidence of normal, branched or swollen cell morphologies upon transient overexpression of AtPIP5K2-EYFP alone, or co-overexpressed either with NtPLC3-RFP or with NtPLC3-RFP H124A in tobacco pollen tubes. Data are from three independent experiments, and > 50 cells were analyzed for each transformation. Asterisks indicate a significant difference to the AtPIP5K2-EYFP control according to a Student’s t-test (*, *p* ≤ 0.05; ***, *p* ≤ 0.01). (**B**) Fluorescence intensities recorded during the co-overexpression of PIP5K2-EYFP with NtPLC3-RFP or with NtPLC3-RFP H124A. Scale bars = 10 µm.

**Figure 5 plants-09-00452-f005:**
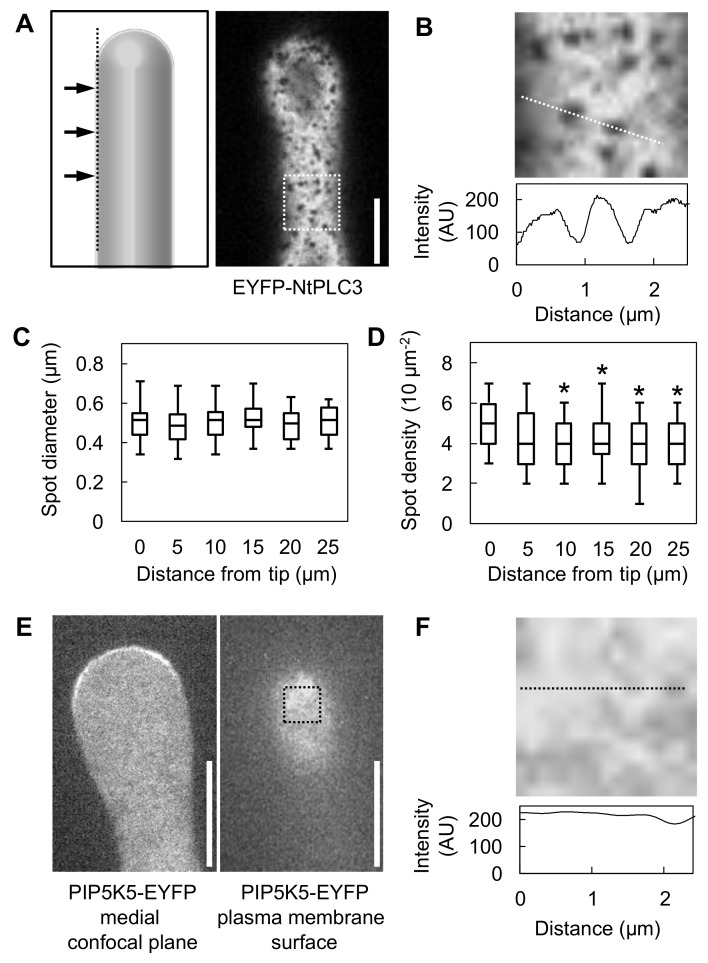
NtPLC3-EYFP localizes in a discontinuous pattern, omitting circular plasma membrane nanodomains. The fluorescence distribution of overexpressed NtPLC3-EYFP in the plasma membrane was assessed by SD microscopy. Only cells displaying low fluorescence intensities were imaged; they exhibited no morphological changes. (**A**) Association of NtPLC3-EYFP with the pollen tube plasma membrane in a discontinuous pattern omitting circular nanodomains, determined by SD microscopy. The pattern is representative for data from three independent experiments, and 15 cells were analyzed. Scale bar = 5 µm. (**B**) Upper panel: Enlargement of the area indicated by the dashed box in (A). Dashed line, trace recorded for intensity analysis. Lower panel: Intensity profile for NtPLC3-EYFP fluorescence along the line. (**C**) Diameter of NtPLC3-EYFP-excluded domains, analyzed at different distances from the pollen tube tip, as indicated. Data are from three independent experiments, and 25 nanodomains were analyzed per distance; (**D**) Area density of NtPLC3-EYFP- excluded domains, analyzed at different distances from the pollen tube tip, as indicated. Data are from three independent experiments and 12 areas were analyzed per distance. (**E**) Association of PIP5K5-EYFP with the pollen tube plasma membrane in a continuous pattern of the plasma membrane, determined by SD microscopy as a control. Left panel, medial confocal section; right panel, peripheral section at the plasma membrane surface. The pattern is representative for data from three independent experiments, and nine cells were analyzed. Scale bar = 5 µm. (**F**) Upper panel: Enlargement of the area indicated by the dashed box in (A). Dashed line, trace recorded for intensity analysis. Lower panel: Intensity profile for PIP5K5-EYFP fluorescence along the line. Asterisks indicate a significant difference from the values at distance zero according to a Student’s t-test (*, *p* ≤ 0.05).

**Figure 6 plants-09-00452-f006:**
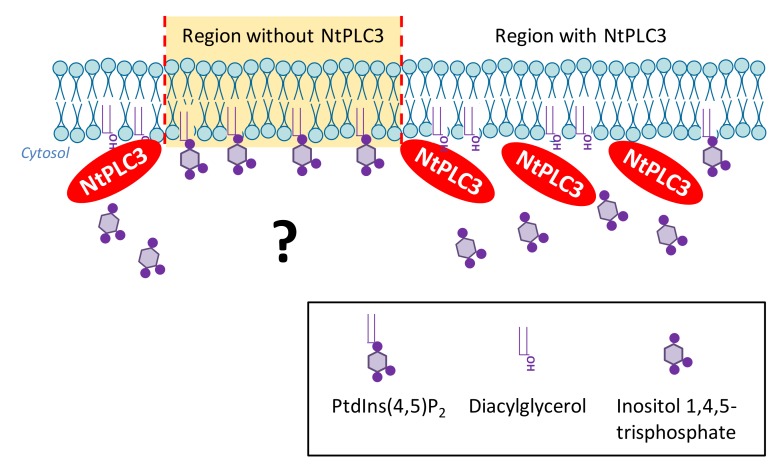
**A simplified model of NtPLC3 antagonizing PtdIns(4,5)P_2_-formation.** A possible role for NtPLC3 is in restricting PtdIns(4,5)P_2_ production in some areas of the plasma membrane while enabling it in other areas. The punctate plasma membrane areas not occupied by NtPLC3-EYFP fluorescence might be sites of enhanced PtdIns(4,5)P_2_ formation and/or residence of PI4P 5-kinases. PtdIns(4,5)P_2_ abundance might thus be controlled and restricted to nanodomains by the action of degrading enzymes, such as PLC. Other explanations are possible.

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
