# Peer review of "Coordinated Localization and Antagonistic Function of NtPLC3 and PI4P 5-Kinases in the Subapical Plasma Membrane of Tobacco Pollen Tubes"

_plants, 2020, doi:10.3390/plants9040452_

Round 1
Reviewer 1 Report
Summary:
In this article, the authors further elaborate on the potential role of NtPLC3 in controlling PtdIns(4,5)P2 turnover at the pollen tubes subapical plasma membrane. Whereas the latter is not necessarily novel, the findings presented here are an added value to the field, and extent/support previous reports. The manuscript is excellently written in terms of content, structure and language.
The simultaneous imaging of PLC3 with PIP5K11 or PIP5K2 is novel. The authors show specific PLC3 subapical plasma membrane localization which might suggest antagonistic breakdown of PtdIns(4,5)P2 in balance with PIP5K-mediated PtdIns(4,5)P2 production. Whereas these results are very informative, it is important to mention that the PLC3 translational fusion constructs which were created here are driven by the constitutively active pollen tube specific LAT52 promoter. Despite this, it is greatly appreciated that the authors show that PLC3-RFP co-expression with PIP5K2-eYFP results in full complementation of the PIP5K2 overexpression phenotype. Nevertheless, the authors need to reflect, albeit in a few sentences, on the possible effects of using an overexpression line to study ‘native’ protein localization.
To further investigate the role of NtPLC3, the authors then characterized pollen tubes expressing a dominant negative version of PLC3. In line with the expected overaccumulation of PtdIns(4,5)P2, both PLC3DN and PIP5K OE lines displayed pollen tube tip swelling. The authors used wild-type pollen tubes for their transient expression. Despite the convincing results presented in Fig. 4, it is still somewhat surprising that the effect of overexpressing PLC3DN in a wild-type background is so strong, since by default native PLC3 is also still expressed (unless there is strong feedback regulation). I’m of the opinion that the authors should at least mention and shortly discuss this in their manuscript, to prevent confusion for future readers.
The true potential of the paper lies in the high spatial resolution imaging of NtPLC3-eYFP plasma membrane localization. I would like to suggest the authors to take this analysis a bit further by quantifying spot diameter and spot density in function of ‘distance to the tip’. The question remains, how do these microdomains localize relative to PIP5K11/2 and where are PtdIns(4,5)P2 domains situated relative to the latter. The authors write that they have put substantial effort into simultaneous imaging of PLC3-RFP and a PtdIns(4,5)P2 biosensor. Knowing the drawbacks of these sensors, these results, even if they are negative, need to be included in the manuscript as they might be of vital importance for future studies. I would like to suggest the authors to include time-lapse colocalization imagery of PIP5Ks and PLC3 in growing pollen tubes using their spinning disc system. They have the tools at their disposal to easily increase the potential impact of this paper.
Minor comments:
Fig 1b,c,d: Please don’t include the brightfield images in the overlays, but keep them as a reference. Restrict the overlays to both fluorescence images (as in Fig. 1a). This is much clearer.
Fig. 1f: A two-channel kymograph would be much more informative, and less prone to speculation (Manually measuring the furthest point to which fluorescence extents along the pollen tube is subjective).
Line 176: Did the authors see a tip branching phenotype in PIP5K5 OE pollen tubes?
Fig. 3a,b: Update labels so that they clearly show that the PLC3 and PIP5K lines are over-expressors.
Line 185: ‘withs’ should be correct to ‘widths’.
Line 178: p > 0.01 should be p ≤ 0.01
Line 192: p > 0.01 should be p ≤ 0.01
Fig. 4a: Add phenotype incidence for untransformed or free RFP expressing PTs as a control.
Line 224: Correct ‘indepoendent’
Line 225-226: ‘asterisks indicate a significant difference to the RFP control, but the phenotype incidence of the RFP control is not shown anywhere.
(C) and (E) should be (a) and (b)
Fig. 5b: depicts x-axis of the intensity graph as µm, not ‘pixels’
Line 298-299: I don’t follow this statement. Isn’t that exactly what you would expect to see if PtdIns(4,5)P2 is regulated by PLC3?
Line 299-301: whilst that might be happening, one would then expect to see phenotypes reflecting PtdIns(4,5)P2 overaccumulation. However, in Fig. 1A, that does not seem to be the case.
Line 301-303: Whilst that is possible, it’s still crucial to share this data with the community.
Line 311: The tendency of pollen tubes to branch depends dramatically on the type of medium used. This might explain why the authors don’t see it in their in vitro growth assays.
Line 338: Please include information on which promoters are driving the PIP5K2 and PIP5K11 translational fusion constructs.
Line 384: ‘wasa’ correct to ‘was’
Line 389: Can the authors specify if and how they tested for normality and homoscedasticity? A student’s t-test is used specifically for normally distributed data. Alternative, one should use non-parametric testing.
Author Response
Reviewer 1
In this article, the authors further elaborate on the potential role of NtPLC3 in controlling PtdIns(4,5)P2 turnover at the pollen tubes subapical plasma membrane. Whereas the latter is not necessarily novel, the findings presented here are an added value to the field, and extent/support previous reports. The manuscript is excellently written in terms of content, structure and language.
The simultaneous imaging of PLC3 with PIP5K11 or PIP5K2 is novel. The authors show specific PLC3 subapical plasma membrane localization which might suggest antagonistic breakdown of PtdIns(4,5)P2 in balance with PIP5K-mediated PtdIns(4,5)P2 production. Whereas these results are very informative, it is important to mention that the PLC3 translational fusion constructs which were created here are driven by the constitutively active pollen tube specific LAT52 promoter. Despite this, it is greatly appreciated that the authors show that PLC3-RFP co-expression with PIP5K2-eYFP results in full complementation of the PIP5K2 overexpression phenotype.
Comment: Nevertheless, the authors need to reflect, albeit in a few sentences, on the possible effects of using an overexpression line to study ‘native’ protein localization.
Response: We thank the reviewer for this suggestion and have added two sentences on the caveats of using overexpression to study the localization of fluorescent fusion proteins in vivo on page 10, lines 303-312.
To further investigate the role of NtPLC3, the authors then characterized pollen tubes expressing a dominant negative version of PLC3. In line with the expected overaccumulation of PtdIns(4,5)P2, both PLC3DN and PIP5K OE lines displayed pollen tube tip swelling. The authors used wild-type pollen tubes for their transient expression. Despite the convincing results presented in Fig. 4, it is still somewhat surprising that the effect of overexpressing PLC3DN in a wild-type background is so strong, since by default native PLC3 is also still expressed (unless there is strong feedback regulation).
Comment: I’m of the opinion that the authors should at least mention and shortly discuss this in their manuscript, to prevent confusion for future readers.
Response: We thank the reviewer for this comment. The results we obtained were very clear and were also consistent with previous findings be Peter Dowd and coworkers (Dowd et al., Plant Cell 2006). The tip-swelling effect was observed upon strong overexpression of PLCDN, possibly accounting for the severe effect. We assume that the overexpression of PLCDN from the Lat52 promoter (see previous point) is strong enough to displace a substantial proportion of the intrinsic PLC3 population present in the wild type cells. An explanatory sentence to this effect has been added in the description of the experiment on page 5, lines 178-180 and 183-185.
The true potential of the paper lies in the high spatial resolution imaging of NtPLC3-eYFP plasma membrane localization.
Comment: I would like to suggest the authors to take this analysis a bit further by quantifying spot diameter and spot density in function of ‘distance to the tip’.
Response: We thank the reviewer for this excellent idea. We have performed these additional analyses and have included the additional data in Figure 5.
The question remains, how do these microdomains localize relative to PIP5K11/2 and where are PtdIns(4,5)P2 domains situated relative to the latter. The authors write that they have put substantial effort into simultaneous imaging of PLC3-RFP and a PtdIns(4,5)P2 biosensor.
Comment: Knowing the drawbacks of these sensors, these results, even if they are negative, need to be included in the manuscript as they might be of vital importance for future studies.
Response: We agree with the reviewer that the data might be helpful. However, as already stated in the original manuscript, our analyses so far did not yield reproducible patterns, which we would feel comfortable to present. We are aware that both reviewers suggested to include the data as they are, but we nonetheless cannot include data that are not of publishable quality. Therefore, we regret that it will not be possible to comply with this request.
Comment: I would like to suggest the authors to include time-lapse colocalization imagery of PIP5Ks and PLC3 in growing pollen tubes using their spinning disc system. They have the tools at their disposal to easily increase the potential impact of this paper.
Response: We fully agree with the reviewer that such data might be helpful. However, we regret to say that we do not in fact have the option to perform dual channel imaging on the spinning disc setup at our disposal. Besides this technical aspect, our access to laboratory facilities is currently severely limited and we have no means to do additional experimental work. As you will be well aware, such limitations will likely apply for an extended period of time. We are asking for your understanding that we cannot easily address this issue and are not providing the data requested. While the additional data would further strengthen our story, we feel they are not essential to illustrate the main points of our study.
Minor comments:
Fig 1b,c,d: Please don’t include the brightfield images in the overlays, but keep them as a reference. Restrict the overlays to both fluorescence images (as in Fig. 1a). This is much clearer.
Response: In all microscopic figure, brightfield layers were removed from the merged images.
Fig. 1f: A two-channel kymograph would be much more informative, and less prone to speculation (Manually measuring the furthest point to which fluorescence extents along the pollen tube is subjective).
Response: We agree that a two-channel kymograph analysis would be more informative. We have performed the kymograph analysis and have included the additional data in the expanded Figure 1. With Fiji, kymographs are obtained by drawing a line along/on top the spot of interest and afterwards the fluorescence intensities, coming from the selected line, are plotted over time. Since pollen tubes exhibit a fast growth rate and their growth is not linear, it was difficult to track the fluorescence intensities values of PLC and PIP5K11 at the cell membrane using kymographs without substantial manual adjustment. To eliminate possible bias and provide an independent analysis of the relative distribution patterns of PLC3 and PIP5K11, we used the machine learning program Ilastik (Berg et al., Nat Methods 16:1226-1232, 2019) to analyze the intensity and distribution of the PIP5K11 and PLC signals; by using the pixel classification and object classification tool, one time frame was used to train the program and afterwards the batch processing function analysed all remaining frames with no further manual processing. The plot (Figure 1E) indicates an unbiased pattern were the areas occupied by PLC and PIP5K11 and their relative positioning over time remain constant.
Line 176: Did the authors see a tip branching phenotype in PIP5K5 OE pollen tubes?
Response: Yes, tip branching was regularly observed in PIP5K5 OE pollen tubes, as previously reported. This information has been added in the description of Figure 3 on page 7, lines 218-220.
Fig. 3a,b: update labels so that they clearly show that the PLC3 and PIP5K lines are over-expressors.
Response: We apologize for the omission and have rephrased the wording in all relevant instances to indicate where proteins were overexpressed.
Line 185: ‘withs’ should be correct to ‘widths’.
Response: Text and figures were rechecked for typographic errors, and these were rectified.
Line 178: p > 0.01 should be p ≤ 0.01
Response: The correct symbol was used in the revised text and figure legends .
Line 192: p > 0.01 should be p ≤ 0.01
Response:The correct symbol was used in the revised text and figure legends .
Fig. 4a: Add phenotype incidence for untransformed or free RFP expressing PTs as a control.
Response: The figure legend was faulty, and we apologize for the mistake. The correct control for this experiment is not RFP, but PIP5K2-EYFP, and the data should be viewed in comparison to the effects of expressed PIP5K2-EYFP. We apologize for the mistake and have corrected the legend. The RFP data are nonetheless present and shown as controls for data in Figure 3C and D. The pollen tube morphology experiments shown in Figure 3C-D and Figure 4A-B were performed side-by-side, so a direct comparison between the experiments is possible.
Line 224: correct ‘indepoendent’
Response: Text and figures were rechecked for typographic errors, and these were rectified.
Line 225-226: ‘asterisks indicate a significant difference to the RFP control, but the phenotype incidence of the RFP control is not shown anywhere.
(C) and (E) should be (a) and (b)
Response: The figure legend was faulty, and we apologize for the mistake. We apologize for the mistake and have corrected the legend. (please see also two comments up)
Fig. 5b: depicts x-axis of the intensity graph as µm, not ‘pixels’
Response: We have changed the axis in Figure 5B to now indicate µm instead of pixels.
Line 298-299: I don’t follow this statement. Isn’t that exactly what you would expect to see if PtdIns(4,5)P2 is regulated by PLC3?
Response: We are not quite sure how to read this comment. The sentence in question was intended to speculate about a contribution of other, non-phosphoinositide lipids to the formation of local membrane areas that differ in their capability to recruit certain proteins, thereby specifying membrane areas from which PLC3 might be excluded. We apologize if this meaning was not clear and have reworded to sentence on page 11, lines 339-341 to now read as follows: "It is possible that local asymmetries in the distribution of certain membrane lipids, such as sphingolipids or sterols, and ensuing differences in the biophysical properties of certain membrane areas may be important aspects of correct recruitment of NtPLC3 to regions where its activity is required."
Line 299-301: whilst that might be happening, one would then expect to see phenotypes reflecting PtdIns(4,5)P2 overaccumulation. However, in Fig. 1A, that does not seem to be the case.
Response: We agree with the reviewer that one might expect a PtdIns(4,5)P2 overaccumulation phenotype in the pollen tube shown in Figure 1A. The intention of the experiment shown in Figure 1A was to see the spatial distribution of EYFP-PIP5K11 and a coexpressed fluorescent biosensor for PtdIns(4,5)P2. Therefore, expression levels in this experiment were kept as low as possible, so no morphological change occurred. At higher expression levels, overproduction of EYFP-PIP5K11 results in massive pollen tube tip-swelling, as was previously reported (Ischebeck et al., Plant J 2011). We have added a sentence on page 3, lines 108-109 to highlight that the expression of constructs in Figure 1 was kept intentionally low to avoid morphological changes.
Line 301-303: Whilst that is possible, it’s still crucial to share this data with the community.
Response: We agree with the reviewer that this information would be great to include. However, as already stated in our response to the equivalent comment by reviewer 1, our analyses so far did not yield reproducible patterns, which we would feel comfortable to present. We cannot include data that are not of publishable quality. Therefore, we regret that it will not be possible to comply with this request.
Line 311: The tendency of pollen tubes to branch depends dramatically on the type of medium used. This might explain why the authors don’t see it in their in vitrogrowth assays.
Response: We are aware of the effects of growth media on the morphology of pollen tubes. Under our conditions, tip branching as that induced by overexpression of PIP5K5 will manifest and has been observed when this protein was overexpressed. This information is now included inb the text on page 7, lines 218-220 (please see also earlier comment above). However, with these positive controls, we still never observed pollen tube tip branching under our conditions, leading us to suggest that media effects can be excluded.
Line 338: please include information on which promoters are driving the PIP5K2 and PIP5K11 translational fusion constructs.
Response: We agree with the reviewer that this information is important to include and we apologize for the omission. Besides the description in the methods section, we have added the requested information in the legends to all figures where overexpression of fusion constructs is shown.
Line 384: ‘wasa’ correct to ‘was’
Response: Done.
Line 389: Can the authors specify if and how they tested for normality and homoscedasticity? A student’s t-test is used specifically for normally distributed data. Alternative, one should use non-parametric testing.
Response: The statistical analysis was done exactly as in previous studies by our and by other groups. The pollen tube morphology data are normal-distributed, as is evident from the median shown for raw data for tip-swelling in Figure 3b. No further statistical analyses were done.
Reviewer 2 Report
The manuscript entitled “NtPLC3 antagonizes and may spatially restrict PtdIns(4,5)P2 in the subapical plasma membrane of tobacco pollen tubes” descripts studies of the interplay between PLC3 and PI4P 5-kinases (PIP5K2, and 11), and finding of the discontinuous plasma membrane localization of PLC3.
The experiments are properly designed and done, and the findings are interesting. However, the title is not accurate or overstated:
- Data in this manuscript only suggests PLC3 antagonizes PI4P 5-kinases rather than antagonizes PtdIns(4,5)P2;
- It is alright to discuss “NtPLC3 may spatially restrict PtdIns(4,5)P2 in the subapical plasma membrane of tobacco pollen tubes” in Discussion section, but it is overstated in the title, as there is no solid data for it.
I would suggest to change the title just based on the solid data.
Besides above, I have some other suggestions for improvement.
- Line 45, “PtdIns(4,5)P2 is formed by PI4P 5-kinases, which are encoded in the Arabidopsis genome by a family of eleven genes”, which genes are evidenced to synthesize PtdIns(4,5)P2 in pollen tube? How about PIP5K2, 5, 11 that are studied here? The authors need to show as much background as possible. This may also be helpful to understand why overexpressing PIP5K5 doesn’t show swelling PTs (line 174’s data).
- To further evidence PLC3 hydrolyzes/restricts PtdIns(4,5)P2 in the plasma membrane, although it is difficult to do co-localization of PLC3 and PtdIns(4,5)P2 for the possible “complementary” nanodomain localization, the authors may test if the hydrolysis product of PtdIns(4,5)P2, InsP3, is significantly increased in the cytosol when PLC3 is overexpressed.
- Need brief definition of subapical region of a pollen tube.
- Line 57-61, “The correct spatial dimensions …. defective patterns of pollen tube expansion [12,13].” hard for understanding.
- What’s the functional difference between PLC1 and PLC3?
- The conclusion for “2.1. NtPLC3 … encompassing domains occupied by PIP5K2 or PIP5K11” is not accurate as PIP5K2-Cherry is mainly in the tube tip with some in the subapical, while PLC3 is mainly in the shank. Please show overlapped images of just red and green channels in Figure 1b for clear co-localization information.
- Figure 1d, also need non-merged images for YFP and RFP channel so that co-localization is clearer. Figure 1f, need error bars.
- All alphabets in figures are lower case letters while capitalized in figure legends.
- Line 154, “When equal amounts (Figure 2A)…” not exactly equal amounts according to Figure 2A. And the control MBP is not in the same gel with others.
- Figure 3a, keep the indicated lines outside a tube rather than inside which covers the image information.
- According to Figure 3d, NtPLC3H124A’s subcellular localization is not changed vs NtPLC3. However, there is no or very low signal of NtPLC3H124A in the membrane in Figure 3a. Which one is correct?
- Does NtPLC3H124A, or OE-PIP5K2, PIP5K11, PIP5K5 affect seed set?
- In Figure 4a, it is not clear what the comparison sets are for stats. In other words, Line 225-226, “RFP control (C) or to the AtPIP5K2-EYFP control (E)” where are the “C” and “E”?
- “diffuse” may not be the correct term to descript the discontinuous localization pattern in section 2.4.
- Where is cf. Figure 1? What does cf mean?
- Briefly introduce the imaging resolution of spinning disc microcopy (why can you get more details by using SD microscopy). This will be helpful for broader readers.
- Figure 5, better to have a negative control: a plasma membrane protein that is not showing punctuated/nanodomain localization. Figure 5c,d, what is the density of YFP signal vs dark dots?
Author Response
Reviewer 2
The manuscript entitled “NtPLC3 antagonizes and may spatially restrict PtdIns(4,5)P2 in the subapical plasma membrane of tobacco pollen tubes” descripts studies of the interplay between PLC3 and PI4P 5-kinases (PIP5K2, and 11), and finding of the discontinuous plasma membrane localization of PLC3.
The experiments are properly designed and done, and the findings are interesting. However, the title is not accurate or overstated:
- Data in this manuscript only suggests PLC3 antagonizes PI4P 5-kinases rather than antagonizes PtdIns(4,5)P2;
- It is alright to discuss “NtPLC3 may spatially restrict PtdIns(4,5)P2 in the subapical plasma membrane of tobacco pollen tubes” in Discussion section, but it is overstated in the title, as there is no solid data for it.
Comment: I would suggest to change the title just based on the solid data.
Response: We apologize if our title was not appropriate. To avoid misleading information in the title, we have rephrased the title to now read: "Coordinated localization and antagonistic function of NtPLC3 and PI4P 5-kinases in the subapical plasma membrane of tobacco pollen tubes".
Besides above, I have some other suggestions for improvement.
Comment:
- Line 45, “PtdIns(4,5)P2 is formed by PI4P 5-kinases, which are encoded in the Arabidopsis genome by a family of eleven genes”, which genes are evidenced to synthesize PtdIns(4,5)P2 in pollen tube? How about PIP5K2, 5, 11 that are studied here? The authors need to show as much background as possible. This may also be helpful to understand why overexpressing PIP5K5 doesn’t show swelling PTs (line 174’s data).
Response:
We agree with the reviewer that the background section should provide all necessary information. We have expanded the introduction on page 2, lines 51-52 as requested and are now listing which PI4P 5-kinase isoforms are known to be expressed and to function in pollen.
Comment:
- To further evidence PLC3 hydrolyzes/restricts PtdIns(4,5)P2 in the plasma membrane, although it is difficult to do co-localization of PLC3 and PtdIns(4,5)P2 for the possible “complementary” nanodomain localization, the authors may test if the hydrolysis product of PtdIns(4,5)P2, InsP3, is significantly increased in the cytosol when PLC3 is overexpressed.
Response:
We appreciate the suggestion, but fear that this is technically not feasible. Such biochemical analysis would require a uniform population of transgenic pollen tubes to be analyzed in comparison with control expressors and nonexpressors. By contrast, our experimental approach is based on the transient expression of our constructs in a population of pollen tubes, with a transformation efficiency of approx. only 0.1 %. This precludes any attempt to do biochemical analysis.
Comment:
- Need brief definition of subapical region of a pollen tube.
Response:
We have added a brief definition of the term on page 2, line 55, as requested.
Comment:
- Line 57-61, “The correct spatial dimensions …. defective patterns of pollen tube expansion [12,13].” hard for understanding.
Response:
We apologize if our wording was hard to follow. We have rephrased the sentence in question to now read "The spatial dimensions of the PtdIns(4,5)P2 domain in the subapical plasma membrane is critical for polar tip-growth, as evidenced by the observation that overproduction of PtdIns(4,5)P2 and ensuing enlargement of the plasma membrane domain occupied by this lipid results in altered patterns of pollen tube expansion."
Comment:
- What’s the functional difference between PLC1 and PLC3?
Response:
Functionally, both enzymes are active PI-PLCs (Dowd et al., Plant Cell 2006; Helling et al., Plant Cell 2006) of both most similar to the PLC zeta-subfamily (Pokotylo et al., Biochimie 2013). The PLC1 mentioned is from Petunia (Dowd et al., Plant Cell 2006), whereas the PLC3 mentioned is from tobacco (Helling et al., Plant Cell 2006). We are not aware of any further differences. This information is now included in the revised text on page 2, lines 77-78.
Comment:
- The conclusion for “2.1. NtPLC3 … encompassing domains occupied by PIP5K2 or PIP5K11” is not accurate as PIP5K2-Cherry is mainly in the tube tip with some in the subapical, while PLC3 is mainly in the shank. Please show overlapped images of just red and green channels in Figure 1b for clear co-localization information.
Response:
We appreciate this comment and have changed the text on page 3, lines 122-124 accordingly to indicate that PIP5K2-Cherry localizes mainly to the tube tip with some signal in the subapical region, while PLC3 localizes mainly to the pollen tube shank. However, our time-lapse analyses and the new kymograph data indicate that the membrane region occupied by PLC3 is larger and frames that occupied by PIP5K2-mCherry. Therefore, we retain the wording "encompassing" in our text. We are presenting this information to illustrate that the localization of PLC3 relative to that of PIP5K2 makes PLC3 a candidate for restricting PIP5K2 effects. Based on the localization data, it is possible that this restriction is focused on the subapical region towards the shank, and we have included this notion in our revised text on page 3, lines 136-139. We have also removed the bright field layer from the merged images, as already mentioned in our response to the corresponding comment by reviewer 1.
Comment:
- Figure 1d, also need non-merged images for YFP and RFP channel so that co-localization is clearer.
Response:
We apologize for the oversight and have added the requested non-merged panels.
Comment:
Figure 1f, need error bars.
Response:
For the time-lapse experiments, we prefer not to compare data between experiments. In particular, we found it very difficult to correct for intrinsic dynamic parameters, such as non-linear growth of the expanding cells. Therefore, we prefer to report a representative experiment. We have nonetheless replaced Figur 1F for another analysis to accomodate the comment on the intrinsic bias associated with manual image processing. The machine-learning based Ilastic analysis is shown in the replacement of Figure 1F and represents an improvement.
Comment:
- All alphabets in figures are lower case letters while capitalized in figure legends.
Response:
We apologize for the oversight and have altered the labeling in the figures to be capitalized as in the legends.
Comment:
- Line 154, “When equal amounts (Figure 2A)…” not exactly equal amounts according to Figure 2A. And the control MBP is not in the same gel with others.
Response:
We agree with the reviewer that not exactly equal amounts of protein were used, as shown in the western. This is not a problem for the interpretation, because the PLC activity is reduced to approx. 1% in the lane with the higher protein amount of the inactive PLC3 H124A. The MBP control was in fact analyzed on the same gel/western, but we have split the gel because of the different protein sizes, so band intensities can be shown side-by-side. This is also not an issue, because the intrinsic PLC activity of maltose-binding protein negative control can be assumed to be zero regardless of the protein amount used.
Comment:
- Figure 3a, keep the indicated lines outside a tube rather than inside which covers the image information.
Response:
We apologize for the line placement and have changed it as requested to no longer interfere with the view of the fluorescence signals.
Comment:
- According to Figure 3d, NtPLC3H124A’s subcellular localization is not changed vs NtPLC3. However, there is no or very low signal of NtPLC3H124A in the membrane in Figure 3a. Which one is correct?
Response:
We thank the reviewer for being so observant and have considered this point. The image shown for PLC3 H124A shows a weaker plasma membrane signal compared to the overall fluorescence than for other cells shown. This pattern is representative and the image does not need to be changed. A possible explanation is the strong overexpression of PLC3 H124A from the Lat52 promoter, which resulted in pollen tube tip swelling (please also see earlier response to comments by reviewer 1 above). The fluorescence intensities shown in Figure 3D do not indicate the plasma membrane signal, but the overall mean fluorescence in a square region of the pollen tube tip, including the plasma membrane. This dataset is provided as a control to illustrate that differences in pollen tube morphologies are not correlated to the overall expression levels of the respective proteins, as indicated by the fluorescence intensities. Importantly, no information about the plasma membrane vs. cytosolic intensity ratios for the individual protein fusion can be gained from these data. Pollen tube tip swelling interferes with the polarized subcellular organization of the pollen tube tip, e.g., allowing the entry of organelles into the "clear zone", making it difficult to compare the intensity of "cytosolic" fluorescence between cells expressing protein fusion which do or which do not influence pollen tube morphology. To address the issue raised by the reviewer, we have added a sentence on page 7, lines 222-224, which indicates that the cytoplasmic fluorescence appears to be higher in pollen tubes displaying tip swelling upon overexpression of PLC3 H124A than was observed in other cells shown.
Comment:
- Does NtPLC3H124A, or OE-PIP5K2, PIP5K11, PIP5K5 affect seed set?
Response:
We agree with the reviewer that this question is relevant to assess the overall physiological relevance of our findings. However, we are a biochemistry/cell biology lab and not set up to investigate seed set or pollen transmission. No experiments along these lines were done or planned, and we ask for your understanding that we feel incapable of providing the requested information. To address this point, we have added a sentence on page 11, lines 372-374 indicating that effects on seed set were not tested.
Comment:
- In Figure 4a, it is not clear what the comparison sets are for stats. In other words, Line 225-226, “RFP control (C) or to the AtPIP5K2-EYFP control (E)” where are the “C” and “E”?
Response:
The figure legend was faulty, and we apologize for the mistake. The correct control for this experiment is not RFP, but PIP5K2-EYFP, and the data should be viewed in comparison to the effects of expressed PIP5K2-EYFP. We apologize for the mistake and have corrected the legend. The RFP data are nonetheless present, but are shown as controls for data in Figure 3C and D. The pollen tube morphology experiments shown in Figure 3C-D and Figure 4A-B were performed side-by-side, so a direct comparison between the experiments is possible. (please see identical response to earlier comment by reviewer 1 above)
Comment:
- “diffuse” may not be the correct term to descript the discontinuous localization pattern in section 2.4.
Response:
We have changed the working to avoid the term "diffuse" to describe the fluorescence distribution of PLC3 as seen by spinning disc microscopy. In the revised text we are now adopting the term "discontinuous" as used by the reviewer.
Comment:
- Where is cf. Figure 1? What does cf mean?
Response:
We apologize if our wording was confusing. "cf." was used to mean "compare figure" to refer to a figure that was previously introduced in the results section. As this was not not helpful, we have removed "cf." and just refer to "Figure 1" (meaning not a special figure, but just Figure 1).
Comment:
- Briefly introduce the imaging resolution of spinning disc microcopy (why can you get more details by using SD microscopy). This will be helpful for broader readers.
Response:
We agree with the reviewer that this information might be helpful to the reader and we have added a short description of the benefits of spinning disc microscopy. However, as this manuscript is not a technical note, we feel that this description must naturally remain short. The description was added on page 13, lines 426-430.
Comment:
- Figure 5, better to have a negative control: a plasma membrane protein that is not showing punctuated/nanodomain localization. Figure 5c,d, what is the density of YFP signal vs dark dots?
Response:
We agree with the reviewer that a negative control would be helpful. We are now providing such a negative control, using the distribution of PIP5K5, an enzyme relevant for our study but apparently not influenced by PLC3. The PIP5K5 marker shows a uniform plasma membrane fluorescence, and this information is now included in the expanded Figure 5.